# Bioengineering of a *Lactococcus lactis* subsp. *lactis* strain enhances nisin production and bioactivity

Roxana Portieles[1]☯, Hongli Xu[1]☯, Feng Chen[1], Jingyao Gao[1], Lihua Du[1], Xiangyou Gao[1], Carlos Borroto Nordelo[2], Qiulin Yue[3], Lin Zhao[3], Nayanci Portal Gonzalez[4], Ramon Santos Bermudez[4], Orlando Borrás-Hidalgo[1,3]*

1 Joint R and D Center of Biotechnology, RETDA, YOTABIO-ENGINEERING CO., LTD., Rizhao, Shandong, P.R. China, 2 VBS Biotec SA, Yuc., México, 3 State Key Laboratory of Biobased Material and Green Papermaking, Shandong Provincial Key Lab of Microbial Engineering, Qilu University of Technology (Shandong Academic of Science), Jinan, People's Republic of China, 4 School of Biological Science and Technology, University of Jinan, Jinan, Shandong, People's Republic of China

☯ These authors contributed equally to this work.
* orlando@yotabio.com

**Data Availability Statement:** All relevant data are within the manuscript and its Supporting Information files.

## Abstract

*Lactococcus lactis* subsp. *lactis* is a food bacterium that has been utilized for decades in food fermentation and the development of high-value industrial goods. Among these, nisin, which is produced by several strains of *L. lactis* subsp. *lactis*, plays a crucial role as a food bio-preservative. The gene expression for nisin synthesis was evaluated using qPCR analysis. Additionally, a series of re-transformations of the strain introducing multiple copies of the *nisA* and *nisRK* genes related to nisin production were developed. The simultaneous expression of *nisA* and *nisZ* genes was used to potentiate the effective inhibition of foodborne pathogens. Furthermore, qPCR analysis indicated that the *nisA* and *nisRK* genes were expressed at low levels in wild-type *L. lactis* subsp. *lactis*. After several re-transformations of the strain with the *nisA* and *nisRK* genes, a high expression of these genes was obtained, contributing to improved nisin production. Also, co-expression of the *nisA* and *nisZ* genes resulted in extremely effective antibacterial action. Hence, this study would provide an approach to enhancing nisin production during industrial processes and antimicrobial activity.

## Introduction

The control of foodborne pathogens constitutes an important issue in the food industry. Diseases produced by these kinds of pathogens are disseminated rapidly [1]. Although diverse actions were deployed to decrease the infections [2], foodborne diseases remain prevalent worldwide [3]. To limit the potential risks, many methods of managing foodborne pathogens have been developed [2]. Bacteriocins can be used as effective alternatives for food preservatives [4, 5].

**Funding:** The authors received no specific funding for this work.

**Competing interests:** The authors have declared that no competing interests exist.

*Lactococcus lactis* is used to produce different fermented milk products. *L. lactis* was previously classified as a *Streptococcus* species. In 1985, it was reclassified as a member of the *Lactococcus* genus [6] and grouped into three subspecies: *L. lactis*, *L. lactis subsp. hordniae*, and *L. lactis subsp. cremoris* [7]. This Gram-positive bacterium is spherical, non-sporulating, homo-lactate, and facultative [8, 9]. *L. lactis* was used in food fermentation for centuries. Also, *L. lactis* produces acid, which helps preserve food. Some strains improve on this preservation feature by producing bacteriocins.

The antimicrobial peptide nisin is composed of 34 amino acids that are produced during the growth of various *L. lactis* strains [10]. Generally, nisin inhibits and prevents the spread of Gram-positive bacteria [11–13]. This constitutes an important element to be used as an excellent food preservative and additive in the food industry [14–18].

Other likely applications of nisin are related to the biomedical field [19–22]. Numerous studies have highlighted the use of nisin in therapeutic approaches. For example, nisin had immunomodulatory effects similar to those human defense peptides [23]. Besides, nisin has been shown to be effective in the treatment of carcinoma [21, 24]. Recently, a solution of nisin Z supplemented with EDTA could be used as antibiotic treatment in diabetic foot infections [22]. An interesting study provided important evidence that nisin Z could be also safely used topically as disinfectant at recommended concentrations without skin toxicity [20].

The sophisticated biosynthesis of nisin has been associated with a group of eleven genes. These genes are grouped into structural genes (*nisA*), transport (*nisT*), post-translational modifications (*nisB*, *nisC*), extracellular precursor processing (*nisP*), immunity (*nisI*, *nisFEG*), and the regulatory genes (*nisR*, *nisK*) [25, 26]. Enhanced nisin yield appears to require a better understanding of the nisin biosynthetic process and its metabolic regulation. The rate and level of nisin production slightly improved when immunity genes were expressed to increase nisin resistance [27].

The analysis of nisin biosynthesis shows the way for carrying out genetic modifications designed to raise nisin yield. The genetic modifications were centralized on the issues that limit the production of nisin. Innovative genetic approaches have increased nisin production with *L. lactis* strains expressing the *aox1*, *pfk13*, and *pkaC* genes [28]. These recombinant nisin producers induced oxidative respiration and glycolytic activity. Additionally, nisin Z production was improved through the use of protoplast fusion and genome shuffle in *L. lactis* ssp. *lactis* YF11 [29]. By simultaneously expressing the *hdeAB*, *Idh*, and *murF* genes, the nisin production of the producer *L. lactis* was improved, as was its acidic tolerance [30]. Herewith, over-expression of the *asnH* gene increased the acidic tolerance in the nisin producer *L. lactis* [31].

Previous research suggests that an increase in nisin production was facilitated by a boosting in the copy number of the crucial genes involved in nisin biosynthesis [32–36]. Additionally, enhancing the copy number of the genes *nisRK* and *nisFEG*, which are responsible for regulating and resisting nisin biosynthesis, significantly increased nisin production [36]. Most of the precedent works were done through a simple genetic transformation and selection of the most promising strains with high production of nisin. However, it is unknown about the impact of the genetic re-transformation on *L. lactis* subsp. *lactis* strains several times with genes involved in nisin synthesis. Previously, the genetic re-transformation was associated with the significant increase of proteins in bacteria and yeast. The re-transformation of *Clostridium acetobutylicum* bacterium with genes expressing stress proteins resulted in a more robust phenotype and improved n-butanol tolerance during fermentations [37]. While the yeast *Pichia pastoris* significantly increased its ability to synthesize insulin when the same approach was used [38].

The use of bacteriocins could be hindered by restricted activity, high concentrations of foodborne pathogens, and antimicrobial resistance. When the bacteriocins are combined or

used in conjunction with other antimicrobial agents, multiple studies have demonstrated additive effects [15, 17, 18]. Thus, this strategy is a good alternative for effective pathogen inhibition. The combination of nisin, carvacrol, and citric acid inhibited Gram-positive and Gram-negative foodborne pathogens [39]. The activity of nisin was enhanced with the addition of EDTA against *L. monocytogenes* and *Escherichia coli* [40]. The combination of polymyxin, colistin, and nisin increased the activity against the biofilm formation of *Pseudomonas aeruginosa* [41]. Also, a *L. lactis* strain that co-expressed nisin and leucocin C demonstrated potent antibacterial activity. This strain had a broad activity range and a good bacteriostatic capacity against Gram-positive foodborne pathogens [42]. Curiously, the biological activity of a *L. lactis* strain co-expressing simultaneously the *nisA* and *nisZ* genes was not previously evaluated.

Therefore, the aim of this study was the characterization and evaluation of serial genetic re-transformations (RT) to enhance the nisin production in the *L. lactis* subsp. *lactis.* Additionally, we constructed a *L. lactis* subsp. *lactis* strain co-expressing the *nisA* and *nisZ* genes with enhanced antimicrobial activity. This work describes how it is possible to achieve a significant increase in nisin production through serial genetic re-transformations in the *L. lactis* subsp. *lactis* strain with high activity.

## Materials and methods

### Bacterial strains, plasmids, and culture conditions

The *Escherichia coli* DH5α (Invitrogen, USA) was grown in an LB (Luria-Bertani) medium (Difco) at 37°C for cloning and the construction of plasmids. The *Lactococcus lactis* subsp. *lactis* CICC 6242 strain used in the experiments was supplied by the China Center of Industrial Culture Collection (Beijing, People's Republic of China). The pGEM-T Easy Vector (Promega, Madison, USA) and pMG36e vector (Addgene, MA, USA) were used in the cloning and recombinant expression experiments. The different fermentations of *L. lactis* subsp. *lactis* were done in 250-mL flasks containing 50 mL of M17 medium (Difco) containing 0.5% of glucose. The *L. lactis* subsp. *lactis* was incubated by shaking at 100 rpm at 30°C for 18 hours without control of pH. The erythromycin (Sangon, Shanghai, China) was used at 250 µg/mL for *E. coli* DH5α and 5 µg/mL for *L. lactis* subsp. *lactis* strains, respectively. The cell growth (O.D. 600), pH and nisin titers (IU/ml) parameters were evaluated every two hours until 18 hours of fermentation.

### Preparation of recombinant strains

Chromosomal DNA from *L. lactis* subsp. *lactis* CICC 6242 strain producing nisin A was extracted using a "Wizard Genomic DNA Purification" kit (Promega, Madison, USA). The *nisA* and *nisRK* genes were amplified through a polymerase chain reaction using the primers listed in Table 1 [36]. The PCRs were performed using the following reagents: 2.5 µL of DNA polymerase buffer, 2.5 µL of dNTPs, 0.5 µL of MgCl$_2$, 0.5 µL of each primer, 0.1 µL of DNA polymerase (Promega, Madison, USA) at 50 ng/µL and enough water in a final volume of 12.5 µL. Amplifications were performed in a Bio-Rad thermocycler T100TM (Bio Rad, USA) with the following program: an initial denaturation step at 94 °C for 5 min, followed by 35 denaturation cycles at 94 °C for 1 min, annealing (*nisA*: 62 °C and *nisRK* 60 °C), extension at 72 °C for 1 min and a final extension step at 72 °C for 7 min. The amplified products were stained with a mix of 2X Blue/Orange Loading Dye (Promega, Madison, USA), loading buffer (1X) and 1X SYBr green Nucleic Acid Gel Stain (Sangon Biotech, Shanghai) dye and were separated by electrophoresis on 1% agarose gels. The sequence integrity from *nisA* and *nisRK* genes were evaluated by sequencing. Herewith, the *nisZ* gene was synthesized (Shanghai Real-Gene Bio-Tech, Inc).

**Table 1. Primers used in the work.**

| Name | Primer sequences (5′- 3′) |
|---|---|
| | **cloning and the construction of plasmids** |
| *nisA*-F | AGAGTCGACCTGCAGGCATGCTAGTACAAAAGATTTTAACTTGGATTT |
| *nisA*-R | AGACTTTGCAAGCTTGCATGCTTATTTGCTTACGTGAATACTACAATGAC |
| *nisRK*-F | CGTAAGCAAATAAGCATGCAAGCTTCCGGCTTTAGGTATAGTGTGT |
| *nisRK*-R | CGTTTTCAGACTTTGCAAAGCTTGTAATCCTTAGAGATTACTAAATTAC |
| | **gene copy number of transformed strains and transcriptional analysis** |
| NisA-qPCR-F | CATCACCACGCATTACAA |
| NisA-qPCR-R | TTTGCTTACGTGAATACTACAA |
| NisB-qPCR-F | TTAGCTTACGGATCTATTCTTG |
| NisB-qPCR-R | CAAATCCACCATATCTTTCTAC |
| NisT-qPCR-F | GCGTCAACTTTCAGGAGG |
| NisT-qPCR-R | GTGCAGCACTTGGTTCATC |
| NisC-qPCR-F | CTTTACATCAAATCGGAGAATC |
| NisC-qPCR-R | CATGTGCTAATCCCATATTCA |
| NisI-qPCR-F | GGGAGAATTGATAAGGATGGT |
| NisI-qPCR-R | ACGGCAAATGCTTCAGTAAGA |
| NisP-qPCR-F | GGAGGGTTTGATAATGAAGAA |
| NisP-qPCR -R | CTGTAATCTGACCTGCGACTT |
| NisR-qPCR-F | GGTATTGGTGGGGATGACTAT |
| NisR-qPCR-R | AACTGCATGTTTATTGCGTTC |
| NisK-qPCR-F | TGGACTATCTTTTGCTCAAGG |
| NisK-qPCR-R | TTAGGATAACTTCTGCCCCAC |
| NisF-qPCR-F | GATGGTATTGCGGAGTTGTTA |
| NisF-qPCR-R | TTATTTCGTGCAACTGATGAC |
| NisE-qPCR-F | TTTCTTATGGGTGGAATACAG |
| NisE-qPCR-R | GCAAACTCATCAAAAGGAATA |
| NisG-qPCR-F | GGATTTCCTTTTGTTCTTTCC |
| NisG-qPCR-R | ATCATTCCTTGTTGCCCTACT |
| 16S-qPCR-F | GATGATACATAGCCGACCTGA |
| 16S-qPCR-R | TTCCCTACTGCTGCCTCC |

The genes were cloned one by one into in pGEM-T Easy Vector (Promega, Madison, USA) and finally in pMG36e vector (Addgene, MA, USA) under p32 promoter to the recombinant expression using PCR Cloning Kit (Qiagen, Germany). The recombinant plasmids were named as pMG36e:*nisA* (*nisA*), pMG36e:*nisZ* (*nisZ*) and pMG36e:*nisRK* (*nisRK*). The recombinant plasmids were first introduced in *E. coli* DH5 cells and cultured on LB agar plates with 250 μg/mL of erythromycin. Restriction enzyme digestions and sequencing analyses were used to examine the DNA plasmids. To obtain the electro-competent *L. lactis* cells, the bacterium was cultivated overnight at 30 ˚C until an O.D.600 of 0.7. The cells were resuspended in 1/100 volume of electroporation solution (0.5 M sucrose and 10% glycerol) and kept on ice, one the cells were washed twice with ice-cold washing solution (0.5 M sucrose and 10% glycerol). The confirmed recombinant plasmids were electro-transformed in the *L. lactis* subsp. *lactis* strain using a Gene Pulser device (Bio Rad, USA) set at 2.2 kV, 200, and 25˚F. Likewise, the transformants pMG36e:*nisA*, pMG36e:*nisZ*, and pMG36e:*nisRK* were selected on M17 medium containing 5 μg/mL erythromycin. The plasmids from the transformant strains were further analyzed by DNA sequencing.

The strategies of re-transformation and expression of *nisA* and *nisRK* genes in the same strain were as follow: a) genetic re-transformation was done in the original *L. lactis* subsp. *lactis* strain with the *nisA* and *nisRK* genes and the colonies were selected by erythromycin resistance and the number of copies of the genes introduced regarding the number of copies of these genes in the original strain; b) independent clones of *L. lactis* subsp. *lactis* resistant to erythromycin and with the highest copies numbers of *nisA* and *nisRK* genes were selected, respectively; c) two more subsequent transformations were carried out using in each step the clones with the highest copies numbers as selection criteria (pMG36e:*nisA* RT and pMG36e:*nisRK* RT), since the antibiotic resistance of the plasmid is the same (S1 Fig).

In order to get both nisins expressed in the same strain, the original and multiple copies of the *nisA L. lactis* subsp. *lactis* strains were also transformed with the *nisZ* gene. The selection of the clones was based on antibiotic resistance, the number of copies of the *nisA* and *nisZ* genes, relative expression, RNA sequencing, and LC-MS/MS analysis (S2 Fig). To test their biological activity, clones expressing nisin A, Z, or both in the same strain were selected. In addition, the genetic stability of expression plasmid vector pMG36e in *L. lactis* subsp. *lactis* was evaluated by culturing transformants pMG36e:*nisA*, pMG36e:*nisA* RT, pMG36e:*nisRK* and pMG36e:*nisRK* RT in 100 µl of GM17 broth without erythromycin selection. The *L. lactis* subsp. *lactis* was maintained at an exponential phase for 120 generations. The *L. lactis* subsp. *lactis* was grown until upper log phase, and 100 µl of culture was inoculated into fresh 100 ml GM17 broth. This method was repeated until the cells had achieved roughly 120 generations. The L. lactis subsp. lactis was then plated on GM17 agar and incubated at 30˚C for 18 hours. A total of 100 colonies were selected at random and subcultured on GM17 agar with erythromycin. Finally, the percentage of colonies that retained the plasmid was calculated.

## Analysis of gene copy number of transformed strains

Total DNA from all the transformed and wild-type *L. lactis* subsp. *lactis* strains were extracted using a "Wizard Genomic DNA Purification" kit (Promega, Madison, USA). Real-time PCR was done using a Rotor-Gene Q machine (Qiagen, Hilden, Germany) with the QuantiTect SYBR Green PCR Kit (Qiagen, Germany). For absolute quantification of the *nisA* and *nisRK* genes in transformed and wild-type *L. lactis subsp. lactis* strains, two standard curves at different dilutions (1:10; 1:100; 1:1000, and 1:10000) were established using the genes previously cloned in pGEM-T Easy Vector (Promega, Madison, USA). The data were determined in terms of copies of the *nisA* and *nisRK* genes / µL of qPCR reaction. The primers used in the experiments are listed in Table 1 [36]. Real-time PCR conditions were as follows: an initial 95˚C denaturation step for 15 min followed by denaturation for 15 sec at 95˚C, annealing for 30 sec at 55˚C, and extension for 30 sec at 72˚C for 40 cycles. Data were analyzed by one-way analysis of variance using GraphPad Prism 5.0 (GraphPad Software, Inc, California). Significant differences among means were determined by Tukey's Multiple Comparison Test least significant difference mean separation at $P < 0.05$.

## Transcriptional analysis by quantitative real-time PCR

The original *L. lactis* subsp. *lactis* and transformed strains were incubated in shaking at 100 rpm at 30˚C during the fermentation. The samples for RNA isolation were taken at 0; 12 and 18 hours of fermentation. Total RNAs were isolated with Trizol (Invitrogen, Carlsbad, CA, USA) according to the manufacturer's instructions. The cDNA was synthesized using the SuperScript III reverse transcriptase kit (Invitrogen, Carlsbad, CA, USA). The primers used in the experiments are listed in Table 1) [36]. For each condition, reactions were done in triplicate in three separate experiments. To standardize cycle threshold (Ct) results, the 16S rRNA

gene was used as an internal control. Quantitative real-time PCR was done using a Rotor-Gene Q machine (Qiagen, Hilden, Germany) with the QuantiTect SYBR Green PCR Kit (Qiagen, Germany). Real-time PCR conditions were as follows: an initial 95˚C denaturation step for 15 min followed by denaturation for 15 sec at 95˚C, annealing for 30 sec at 55˚C, and extension for 30 sec at 72˚C for 40 cycles. The relative levels of genes were expressed as 'mean normalized expression' data using Q-Gene software [43]. The Q-Gene software application is a tool for dealing with complex quantitative real-time PCR experiments on a large scale, significantly accelerating and rationalizing experimental setup, data analysis, and data management while ensuring maximum reproducibility. Data were analyzed by one-way analysis of variance using GraphPad Prism 5.0 (GraphPad Software, Inc, California). Significant differences among means were determined by Tukey's Multiple Comparison Test least significant difference mean separation at $P < 0.05$.

## Nisin quantification and bioactivity assay

The nutrient broth (NB) medium (peptones 10 g/L, beef extract 1 g/L, yeast extract 2 g/L, sodium chloride 5 g/L, and pH 6.8) for *Micrococcus luteus* was used during the bioassay agar plates [44]. Before boiling and sterilizing, 0.75% Bacto agar (Difco) and 1% of Tween 20 were added to the medium composition. After autoclaving, the medium was inoculated with 1% of the 24-h culture of *M. luteus*. The concentration of the microorganism was approximately $10^8$ colony-forming units/mL. The bioassay agar was placed aseptically into sterile Petri dishes (100 X 15 mm) and allowed to solidify for 3 hours. Four holes were done on each plate using a 7-mm outer diameter stainless steel borer. A well was filled with an aliquot (100 μL) of standard nisin solution and samples, and the bioassay agar plates were incubated at 30 ˚C for 24 hours. The diameter of the inhibitory zone was measured in the nisin-sensitive microorganism. Three replication were used in the bioassay. Multiple comparisons using Tukey's test were done.

Also, the nisin quantification was estimated by agar diffusion activity assays and expressed in international units / mL (IU/mL) [44]. A stock nisin solution (1,000 IU/mL) was prepared by adding 0.025 g of commercial nisin$10^6$ IU/g (Sigma Aldrich) into 25 mL of the sterile diluent solution of 0.02 N HCl: *L. lactis* fermentation medium (9:1 by volume), which consisted of (per L): 80 g of glucose, 10 g of peptone, 10 g of yeast extract, 10 g of $KH_2PO_4$, 2 g of NaCl, and 0.2 g of $MgSO_4.7H_2O$. Standard nisin solutions of 500, 400, 300, 200, 100, 50, 25, 10, 5, and 0 IU/mL were prepared using the 1,000 IU/mL nisin stock solution to construct the standard curve. To establish a standard curve, the diameters of the inhibition zones were plotted against the log10 concentrations of nisin. A line regression equation was determined for a standard curve. The diameter of the inhibition zone obtained from the nisin standard solution was correlated with the bioassay of sensitivity [44].

## Chloroform extraction and LC-MS/MS analysis

The transformed *L. lactis* subsp. *lactis* and wild-type strains overnight culture were inoculated in M17 medium (Difco) and incubated in shaking at 100 rpm at 30˚C for 18 hours. Bacterium cells were pelleted at 7500 g for 15 min in a refrigerated centrifuge (12 ˚C) and the supernatant was collected. Chloroform was added to the supernatant (1:1), stirred vigorously for 20 min, and centrifuged at 10400 g (12 ˚C) for 30 min. The precipitated and interface solids were collected. The solids were dried in a chemical hood overnight. Finally, the solids were suspended in 1 ml of Tris buffer (0.1 mol/L, 5±10 ml, pH 7.0) with agitation overnight at 8 ˚C. A membrane filter (0.20 m) was used to filter the extraction, which was suspended in 1 ml of Tris buffer. The filtered sample solution was then used for LC-MS/MS analysis. An HPLC system

(1200 series, Agilent, Santa Clara, CA) and triple quadruple mass spectrometer (Agilent 6410), were used to identify and quantify the analyses. A CW-C18 column (50x2 mm, 3 m; Imtakt, Portland, OR) was used for the chromatographic analysis, and the column oven temperature was maintained at 40 ˚C. The determination of nisin A and nisin Z content in the fermentation was done quantitatively using LC-MS/MS analysis with positive mode electrospray ionization and the multiple reaction-monitoring parameters. Data were analyzed with Agilent 6410 Quantitative analysis version (B.02.01) analyst data-processing software.

## Results

### Characterization of *L. lactis* subsp. *lactis* wild-type strain

The fermentation profiles of the *L. lactis* subsp. *lactis* wild-type strain were determined at different time points. The nisin titer was 324 IU/ml after 18 hours of culture. Further, the cell growth reached a maximum of 1.81 optical density after 12 hours. Additionally, there was a strong reduction in pH which had a value of 4.5 upon finishing the fermentation process (Fig 1).

Also, the expression of the genes related to nisin production in the wild-type strain of *L. lactis* subsp. *lactis* was evaluated. The majority of the nisin biosynthetic genes had variable transcription levels. The *nisI*, *nisF*, and *nisG* genes showed higher levels of expression. However, the *nisR* and *nisK* genes had low expression in the different time points analyzed. Also, the *nisA* gene showed reduced expression (Fig 2). According to these results, the genes involved in nisin pre-peptide synthesis (*nisA*) and two-component signal transduction (*nisRK*) were selected for further genetic transformation in the *L. lactis* subsp. *lactis* wild-type strain.

### Serial genetic re-transformations to nisin production

We designed an approach based on a re-transformation of the *nisA* and *nisRK* genes into the wild-type *L. lactis* subsp. *lactis* strain. The transcription levels of *nisA* and *nisRK* genes were higher in modified *L. lactis* subsp. *lactis* strains generated following a single transformation than in the original strain (Fig 3). However, the relative expression was significantly higher in re-transformed *L. lactis* subsp. *lactis* strains compared with the single transformed and wild-type strains, respectively (Fig 3).

The growth of the engineered *L. lactis* subsp. *lactis* strains was determined through their optical densities at 600 nm. The engineered *L. lactis* subsp. *lactis* pMG36e:*nisA*, pMG36e:*nisA* RT, pMG36e:*nisRK*, and pMG36e:*nisRK* RT strains displayed a similar cell growth density compared with the wild-type strain. There was no effect of the over-expression of these genes on cell growth (Fig 4A). A high and significant number of copies of the *nisA* genes were obtained in the engineered pMG36e:*nisA* RT and pMG36e:*nisRK* RT strains compared with the single transformed and wild-type strains, respectively (Fig 4B).

The nisin productivity of the original strain and the engineered strains (pMG36e:*nisA*, pMG36e:*nisA* RT, pMG36e:*nisRK*, and pMG36e:*nisRK* RT) were evaluated in shake flasks after 18 h using agar diffusion activity assays. The wild-type and transformed strains after single and three-fold transformations with empty plasmid pMG36e exhibited similar nisin production, suggesting this had no effect on the nisin production. Though, the pMG36e:*nisA* and pMG36e:*nisRK* strains showed a high production of nisin, with 442.5 IU/ml and 369.2 IU/ml, respectively, after 18 hours, compared with the wild-type strain (169.3 IU/ml) (Fig 4C).

Among these engineered strains, pMG36e:*nisA* RT (1111.5 IU/ml) and pMG36e:*nisRK* RT (1041.5 IU/ml) had the greatest increments in nisin production (Fig 4C). These results stated that the strains genetically modified through re-transformation had a positive effect on nisin yield. Also, with or without erythromycin selection pressure, the pMG36e genetically stable

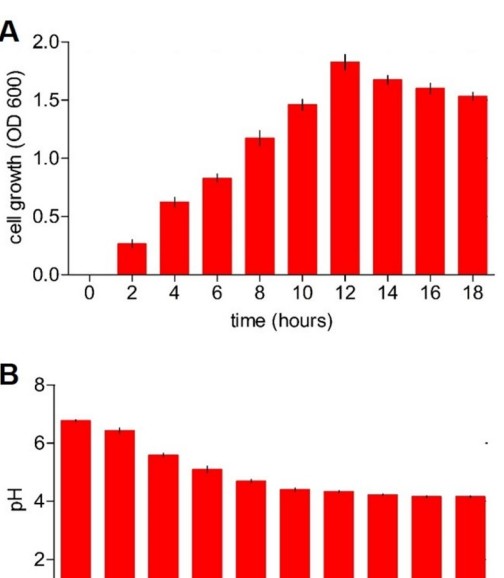

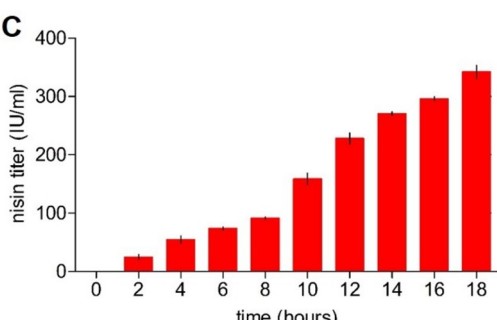

**Fig 1. Evaluation of different parameters during the fermentation of the *L. lactis* subsp. *lactis* strain wild type.** (A) Temporal growth profile. (B) pH of the culture. (C) Nisin A production. The bars represent means of three independent replicates ± standard error ($P<0.05$).

percentage of retention in *L. lactis* subsp. *lactis* was about 98 percent after 120 generations of continuous incubation (S3 Fig).

## Co-expressing *nisA* and *nisZ* genes

To evaluate the advantages of the co-expression of *nisA* and *nisZ* genes in terms of biological activity, we compared them with the original strain. Antimicrobial assays were done with engineered *L. lactis* subsp. *lactis* pMG36e:*nisA*, *pMG36e:nisZ*, and pMG36e:*nisA* + pMG36e:*nisZ* constructs. The *L. lactis* subsp. *lactis* strain producing nisin A and nisin Z was verified by RNA sequencing and mass spectrometry, respectively. For biological activity, a strain selected with similar transcript expression levels from the *nisA* and *nisZ* genes was used. Additionally, LC-MS/MS analysis showed the production of nisin A and Z in the same selected strain at similar concentrations. All the strains inhibited the *M. luteus* indicator strain through antagonistic bioactivity assays. However, the strain expressing the *nisA* and *nisZ* genes had the highest

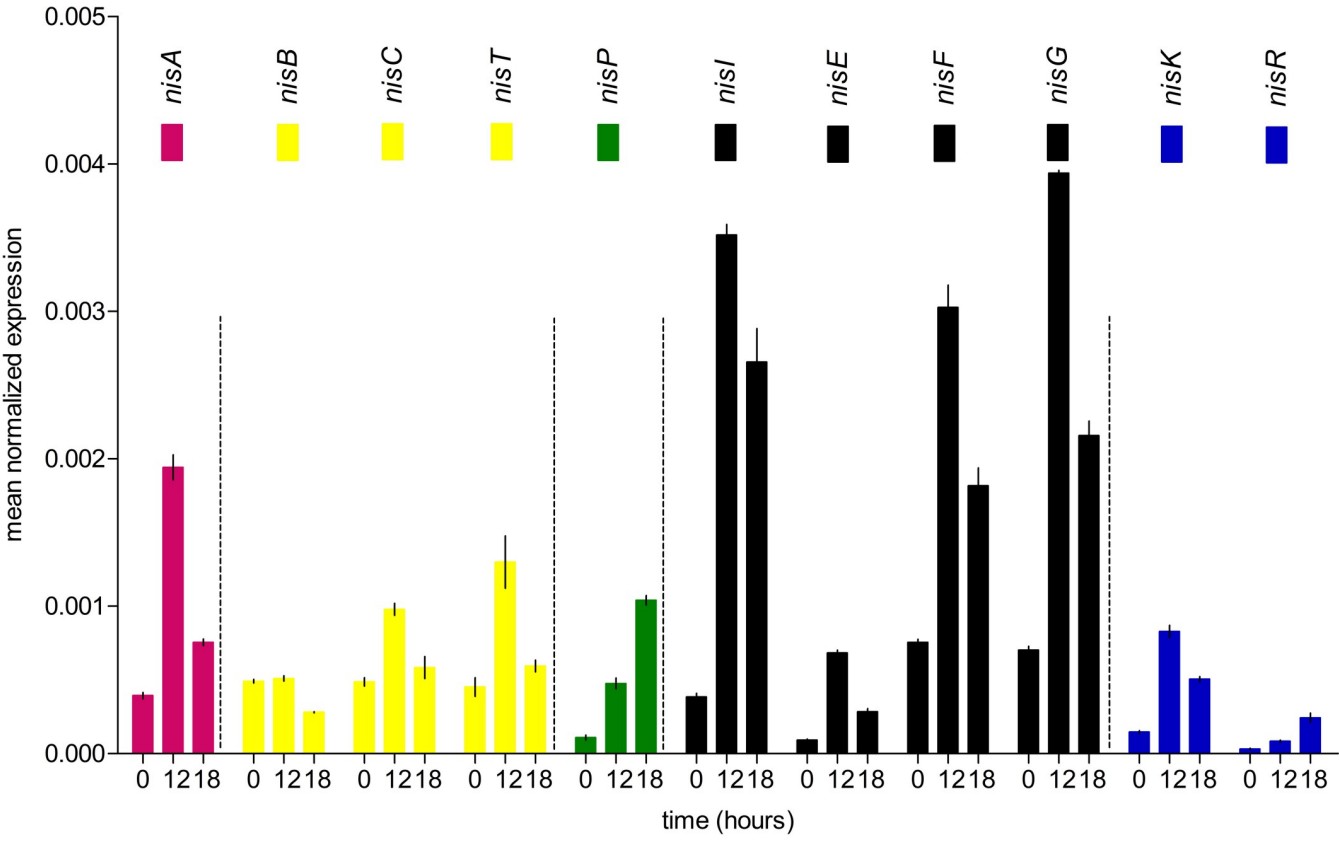

**Fig 2. Transcriptional analysis of genes involved in the biosynthesis of nisin in the *L. lactis* subsp. *lactis* strain using qRT-PCR.** For each condition, reactions were done in triplicate in three separate experiments. To standardize cycle threshold (Ct) results, the 16S rRNA gene was used as an internal control. The relative expression was expressed as 'mean normalized expression' data using Q-Gene software [43]. The bars represent means of three independent replicates ± standard error (P<0.05).

inhibitory effect on the indicator strain (Fig 5). In terms of bacteriostatic activity against *M. luteus*, co-expression of nisin A and Z was superior to the original strain.

## Discussion

The parameters of cell growth and pH during fermentation remained adequate in the wild-type *L. lactis* subsp. *lactis* strain. A reduction in pH typically associated with lactic acid production was observed at the end of fermentation, which is closely related to nisin production. Although the wild-strain of *L. lactis* subsp. *lactis* produced certain levels of nisin at the end of fermentation by shake flask culture, the levels were low compared to previous work where other wild-strain of *L. lactis* subsp. *lactis* were used [36, 45, 46]. To look for the causes of these low levels, a molecular characterization through qPCR was developed with the aim of determining how the set of genes involved in nisin production were expressed.

Different approaches for understanding nisin biosynthesis and its metabolism have been evaluated [25, 46–48]. It is important to note that each of these genes should not be analyzed in a separate context. However, the *nisR* and *nisK* gene expression was significantly low in the wild-type *L. lactis* subsp. *lactis* strain. The *nisA* and *nisRK* genes display a significant role in the synthesis and regulation of nisin [36, 45, 49]. It was evident that the expression of these genes was limited, which could explain in some way the low production of nisin obtained. This has been demonstrated in other strains of *L. lactis* subsp. *lactis* how the increasing of these genes

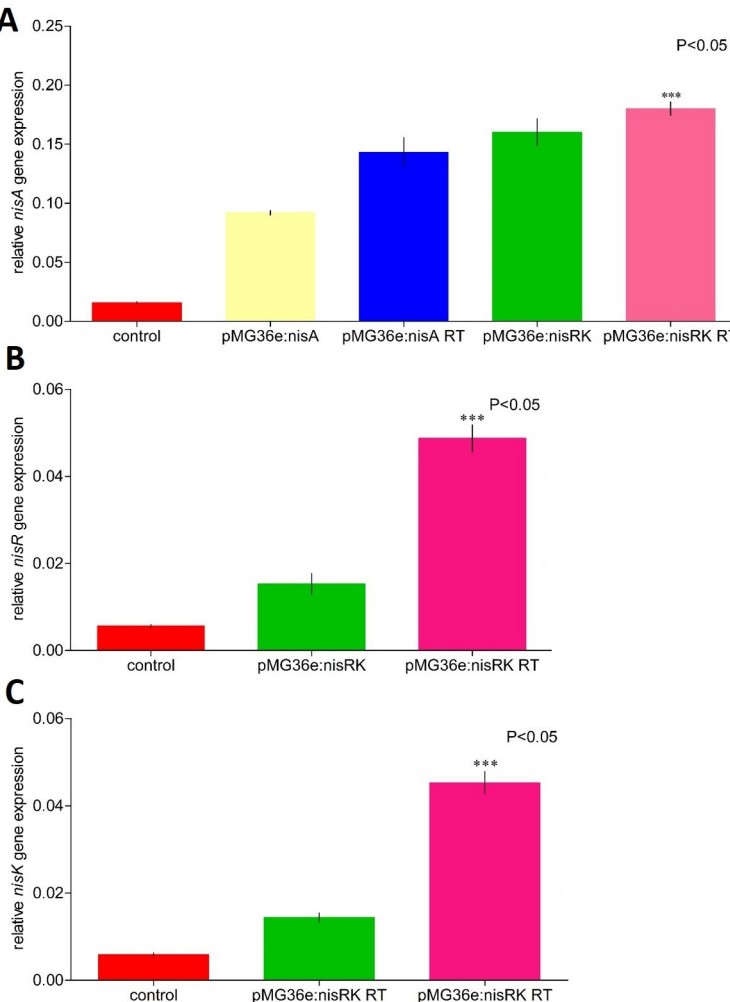

**Fig 3. Transcriptional analysis of the *nisA*, *nisR*, and *nisK* genes in the wild-type and engineered *L. lactis* subsp. *lactis* strains using qRT-PCR.** The relative expression values were determined regarding the 16S rRNA gene used as reference internal gene. The relative levels of genes were expressed as 'mean normalized expression' data using Q-Gene software [43]. The bars represent means of three independent replicates ± standard error (P<0.05).

could enhance nisin production [46]. Specifically, when the produced nisin induces the two-component signal transduction, the *nisRK* genes display an important role [25, 26, 47]. Although, we focused on these genes. Others genes involved in the nisin biosynthetic could be evaluated in the future.

Given this, increasing the expression of these genes could, in principle, improve the production of nisin in the wild-type strain. Consequently, we increased the expression of these genes by introducing multiple copies of the *nisA* and *nisRK* genes in *L. lactis* subsp. *lactis*. To increase the expression of nisin, the *nisA* and *nisRK* genes were cloned in a high-copy-number vector. The resulting plasmids, pMG36e:*nisA* and pMG36e:*nisRK*, were introduced into *L. lactis* subsp. *lactis* through single- and three-fold genetic re-transformation. The results showed how it is possible to first increase gene expression and then increase the production of nisin. The highest expression profiles were detected in the engineered strains obtained through the re-transformation of the wild-type strain compared with the single transformation wild-type strain. Our data support the idea that high expression levels of the *nisA* and *nisRK* genes in the

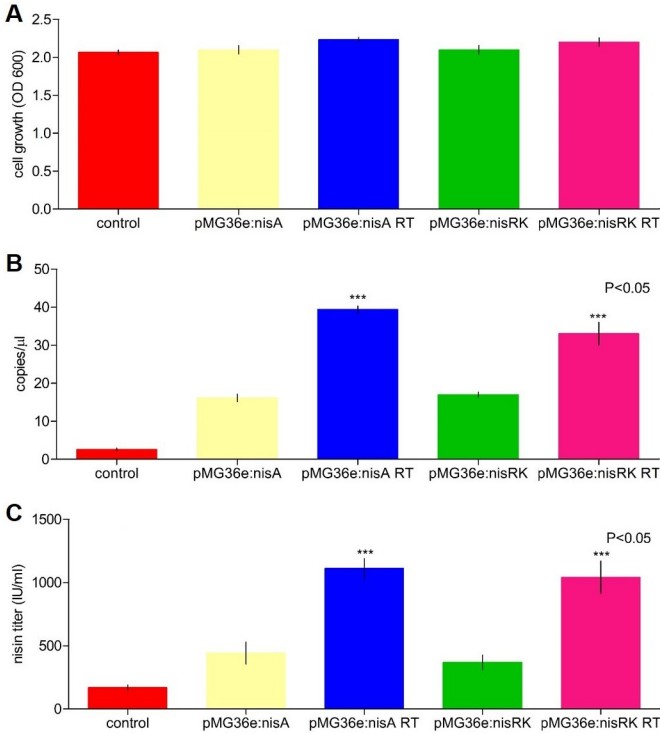

**Fig 4. Characterization of the *L. lactis* subsp. *lactis* wild-type and engineered strains.** (A) cell growth was evaluated through of optical density OD600 after 18 hours of fermentation. (B) the value data were determined in terms of copies of the *nisA* and *nisRK* genes / µL of qPCR reaction. (C) the nisin A quantification was determined by activity assay in the *L. lactis* subsp. *lactis* wild-type and engineered strains. The bars represent means of three independent replicates ± standard error (P<0.05).

engineered strains increase nisin production. We successfully achieved a high production of

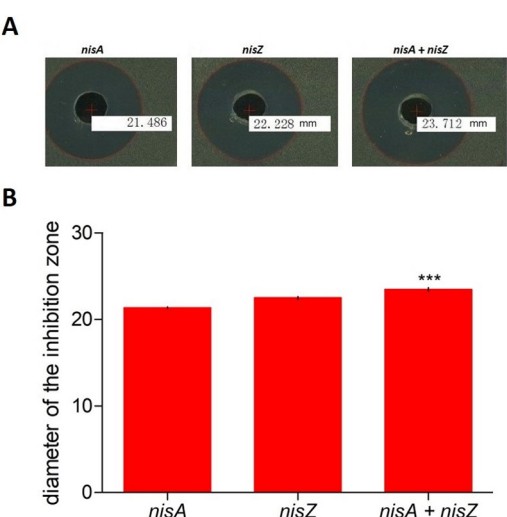

**Fig 5. Antimicrobial activity of *L. lactis* subsp. *lactis* strains expressing *nisA*, *nisZ* and *nisA* + *nisZ* genes against *M. luteus*.** The plates' holes were filled with 150 ml of pasteurized supernatant from overnight cultures of *L. lactis* subsp. *lactis* strains. (A) Inhibition area; (B) inhibition diameter in millimeters. The bars represent means of three independent replicates ± standard error (P<0.05).

nisin without any influence over the growth and stability of the engineered *L. lactis* subsp. *lactis* strains in the shake flasks. Interesting, in *L. lactis* subsp. *lactis*, pMG36e demonstrated high genetic stability. The genetic stability of pMG36e was not affected by erythromycin selection pressure. Previously, the stability of the pMG36e plasmid was demonstrated in the *L. lactis* M4 strain [50].

On the other hand, the cell growth of high-yield engineered strains showed no differences compared with the control strains. There was no toxicity effect produced by the presence of multiple copies of these genes, including the engineered strains obtained by re-transformation. The results provide an approach for improving nisin production. Additional studies related to the optimization of fermentation conditions in engineered strains in scale production should be developed.

Diverse efforts have been developed to increase nisin activity [42, 51, 52]. Combinations with other bacteriocins have allowed an increase in activity [42, 52]. Additionally, the combination with compounds that weaken the cell wall of Gram-negative bacteria such as *E. coli* has increased the spectrum of action [42, 53, 54]. In this context, the differences between nisin A and nisin Z are minimal [14]. Within these differences, the different amino acids in position 27 and the ability of nisin Z to achieve a higher level of inhibition with respect to nisin A, mainly due to its ability to have a greater degree of diffusion in agar with respect to nisin A, are the most important [14].

An interesting point was how *L. lactis* subsp. *lactis* would be expressing nisin A and nisin Z simultaneously. The results suggest that there is a potentiation of activity when the two nisins are produced in the same strain of *L. lactis* subsp. *lactis*, compared to those that are expressed separately. There was a positive effect between them, which contributes to better activity in terms of inhibition activity.

Several analyses have shown that nisin has additive or synergistic effects when used with other antimicrobial compounds [15, 17, 18, 40, 41]. However, there was no evidence available about the antibacterial activity of the *L. lactis* subsp. *lactis* strain co-expressing nisin A and nisin Z. Our findings show that the co-expression of these two bacteriocins with minor variations resulted in extremely effective antibacterial action against *M. luteus*.

Finally, many efforts have been made to seek strategies that allow an increase in the production of nisin. Within these, the increase in the expression of genes related to the biosynthesis of nisin was extensively addressed. Our work, although focused on this issue, introduces the strategy of genetic retransformation for an increase in the number of significant copies, without affecting stability and with a marked increase in the yield of nisin. On the other hand, it was interesting to evaluate the biological effect of the expression of the *nisA* and *nisZ* genes in the same strain of *L. lactis*. It is well known for the amino acid difference at position 27 and the greater agar diffusion capacity of nisin Z compared to nisin A. This was demonstrated; however, it was curious to observe how this strain, which produced the same concentration of nisin A and Z, had a greater inhibitory effect. Further studies will be necessary to understand the reason for this increase in activity and behavior during scale-up in fermenters. Additionally, optimization in fermenter conditions would be important to introduce these *L. lactis* strains genetically modified in a production flow.

## Supporting information

**S1 Fig. Schematic representation of the strategies of re-transformation and expression of *nisA* and *nisRK* genes in *L. lactis*.**
(TIF)

**S2 Fig. Schematic representation of the strategies of co-expression of *nisA* and *nisZ* genes in *L. lactis*.**
(TIF)

**S3 Fig. Evaluation of genetic stability of the pMG36e plasmid expressing of *nisA* and *nisRK* genes in *L. lactis*.**
(TIF)

## Author Contributions

**Conceptualization:** Roxana Portieles, Hongli Xu, Orlando Borrás-Hidalgo.

**Data curation:** Roxana Portieles, Hongli Xu, Qiulin Yue, Lin Zhao, Ramon Santos Bermudez, Orlando Borrás-Hidalgo.

**Formal analysis:** Roxana Portieles, Hongli Xu, Jingyao Gao, Qiulin Yue, Lin Zhao, Ramon Santos Bermudez, Orlando Borrás-Hidalgo.

**Funding acquisition:** Hongli Xu, Orlando Borrás-Hidalgo.

**Investigation:** Roxana Portieles, Hongli Xu, Feng Chen, Jingyao Gao, Lihua Du, Carlos Borroto Nordelo, Qiulin Yue, Lin Zhao, Nayanci Portal Gonzalez, Ramon Santos Bermudez, Orlando Borrás-Hidalgo.

**Methodology:** Roxana Portieles, Hongli Xu, Lihua Du, Carlos Borroto Nordelo, Lin Zhao, Nayanci Portal Gonzalez, Orlando Borrás-Hidalgo.

**Project administration:** Hongli Xu, Xiangyou Gao, Carlos Borroto Nordelo.

**Resources:** Hongli Xu, Carlos Borroto Nordelo, Orlando Borrás-Hidalgo.

**Software:** Lin Zhao, Ramon Santos Bermudez, Orlando Borrás-Hidalgo.

**Supervision:** Hongli Xu, Xiangyou Gao, Carlos Borroto Nordelo, Orlando Borrás-Hidalgo.

**Validation:** Roxana Portieles, Hongli Xu, Feng Chen, Xiangyou Gao, Nayanci Portal Gonzalez, Ramon Santos Bermudez, Orlando Borrás-Hidalgo.

**Visualization:** Lihua Du, Xiangyou Gao, Carlos Borroto Nordelo, Ramon Santos Bermudez, Orlando Borrás-Hidalgo.

**Writing – original draft:** Orlando Borrás-Hidalgo.

**Writing – review & editing:** Orlando Borrás-Hidalgo.

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
