## [Decision Letter · Decision Letter 0]

27 Feb 2023

PONE-D-23-01318Bioengineering of a Lactococcus lactis subsp. lactis strain enhances nisin production and bioactivity.PLOS ONE

Dear Dr. Orlando Borras-Hidalgo,

Thank you for submitting your manuscript to PLOS ONE. After careful consideration, we feel that it has merit but does not fully meet PLOS ONE’s publication criteria as it currently stands. Therefore, we invite you to submit a revised version of the manuscript that addresses the points raised during the review process.

We look forward to receiving your revised manuscript.

Kind regards,

Awatif Abid Al-Judaibi, PhD

Academic Editor

PLOS ONE

Journal Requirements:

"the funders had no role in study design, data collection and analysis, decision to publish, or preparation of the manuscript."

4. Thank you for stating the following in the Financial Disclosure section: 

"the funders had no role in study design, data collection and analysis, decision to publish, or preparation of the manuscript."

We note that one or more of the authors are employed by a commercial company: YOTABIO-ENGINEERING CO., LTD

Please respond by return email with an updated Funding Statement and Competing Interests Statement and we will change the online submission form on your behalf.

Reviewers' comments:

Reviewer's Responses to Questions

**Comments to the Author**

1. Is the manuscript technically sound, and do the data support the conclusions?

Reviewer #1: Yes

Reviewer #2: Yes

2. Has the statistical analysis been performed appropriately and rigorously? 

Reviewer #1: Yes

Reviewer #2: Yes

3. Have the authors made all data underlying the findings in their manuscript fully available?

Reviewer #1: Yes

Reviewer #2: Yes

4. Is the manuscript presented in an intelligible fashion and written in standard English?

Reviewer #1: Yes

Reviewer #2: Yes

5. Review Comments to the Author

Reviewer #1: General comments:

The study explores the genetic transformation of L. lactis subsp. lactis wild-type strain to increase nisin production. Specifically, the study focuses on the overexpression of the nisA and nisRK genes in L. lactis subsp. lactis wild-type strain and evaluates the impact of these transformations on cell growth, nisin production, and gene expression. The study also compares the nisin productivity of the original strain and the engineered strains, including those that underwent serial genetic re-transformations. The study also shows that the co-expression of these two bacteriocins results in extremely effective antibacterial action against M. luteus. Overall, the findings of the study demonstrate the potential of genetic engineering to enhance the production of nisin and provide insights into the regulation of the genes involved in nisin biosynthesis. In general, quality of paper in terms of study design, and discussion of the results is fairly good.

Below are my comments and suggestions to help improve the quality of this manuscript before it could be considered for publication.

1. The study provides valuable insights into the culture conditions, preparation of recombinant strains, and gene expression strategies for the recombinant expression of nisin in Lactococcus lactis subsp. lactis. The study could be useful for further optimization of the expression of nisin and other antimicrobial peptides in Lactococcus lactis.

2. This study focused on increasing the expression of the nisA and nisRK genes to increase nisin production, but there may be other genes involved in nisin biosynthesis that could be targeted to further enhance production. Additional studies could investigate other genes involved in nisin production and their potential for increasing production.

3. There are few studies indicating the potential toxicity of nisin for topical use such as “In vitro assessment of skin sensitization, irritability and toxicity of bacteriocins and reuterin for possible topical applications”. Therefore authors should explore if there is any research/concern regarding the potential toxicity or allergenicity of nisin in human.

4. This study showed that co-expressing nisin A and nisin Z resulted in a potentiation of activity against M. luteus only. It would be interesting to conduct additional antibacterial tests to evaluate the activity against other bacterial strains.

5. Please update the paper with latest references. There is only one reference from year 2022, while no reference from year 2023.

Reviewer #2: Nice presentation. Good and ineresting piece of work. the is a very significant piece of work. these type of work must be published, tablesand figure are well presented. usefull inovative work with industrial application.

6. PLOS authors have the option to publish the peer review history of their article (what does this mean?). If published, this will include your full peer review and any attached files.

Reviewer #1: **Yes: **Sidra Abbas

Reviewer #2: No

---

## [Author Response · Author response to Decision Letter 0]

8 Mar 2023

Dear Prof. Dr. Awatif Abid Al-Judaibi, 

Academic Editor

PLOS ONE

Thank you for your consideration of our manuscript entitled "Bioengineering of a Lactococcus lactis subsp. lactis strain enhances nisin production and bioactivity." (PONE-D-23-01318). We have reviewed the comments provided by the reviewers and have thoroughly revised the manuscript. We found the comments helpful and believe that our revised manuscript represents a significant improvement over our initial submission. We would like to thank you and the referees for the excellent comments made to our manuscript, and we feel that the comments helped us obtain an improved version of our manuscript.

We have attempted to address all comments from the reviewers. Enclosed is a revised version of the manuscript with all changes indicated in red font to facilitate the review process.

Best regards,

Prof. Dr. Orlando Borrás-Hidalgo

 

Reviewers' comments:

Reviewer #1: 

General comments:

The study explores the genetic transformation of L. lactis subsp. lactis wild-type strain to increase nisin production. Specifically, the study focuses on the overexpression of the nisA and nisRK genes in L. lactis subsp. lactis wild-type strain and evaluates the impact of these transformations on cell growth, nisin production, and gene expression. The study also compares the nisin productivity of the original strain and the engineered strains, including those that underwent serial genetic re-transformations. The study also shows that the co-expression of these two bacteriocins results in extremely effective antibacterial action against M. luteus. Overall, the findings of the study demonstrate the potential of genetic engineering to enhance the production of nisin and provide insights into the regulation of the genes involved in nisin biosynthesis. In general, quality of paper in terms of study design, and discussion of the results is fairly good.

Below are my comments and suggestions to help improve the quality of this manuscript before it could be considered for publication.

Authors: We appreciated very much the opinion of the reviewer about the topic of our work. We are so thankful for the excellent revision and suggestions. These enhance significantly the quality of our manuscript.

1. The study provides valuable insights into the culture conditions, preparation of recombinant strains, and gene expression strategies for the recombinant expression of nisin in Lactococcus lactis subsp. lactis. The study could be useful for further optimization of the expression of nisin and other antimicrobial peptides in Lactococcus lactis.

Authors: Thank for your suggestion. We are agreeing with the reviewer. Further optimization will be necessary to scale-up the use of these strains. We have included this information suggested by the reviewer in “Discussion” section as follows:

“Additionally, optimization in fermenter conditions would be important to introduce these L. lactis strains genetically modified in a production flow.”

2. This study focused on increasing the expression of the nisA and nisRK genes to increase nisin production, but there may be other genes involved in nisin biosynthesis that could be targeted to further enhance production. Additional studies could investigate other genes involved in nisin production and their potential for increasing production.

Authors: We are agreeing with the reviewer. There are an important group of genes involved in nisin biosynthesis that were not studied. The nisin biosynthesis is very complex, where eleven genes are involved. These genes have a key role in the different biosynthesis stage. Sometimes, it is not easy produce high quantities of nisin, because the bacterium itself regulate the expression to avoid the toxicity. So, in our case, the strain that we were using in the experiments had a low expression of nisA and nisRK genes, when the expression was evaluated by qPCR. That was the reason why we focused out aim in these genes. However, we agree with the reviewer that, other genes should be evaluated in the future. We have included this information suggested by the reviewer in “Discussion” section as follows:

“Although, we focused on these genes. Other genes involved in the nisin biosynthetic could be evaluated in the future.”

3. There are few studies indicating the potential toxicity of nisin for topical use such as “In vitro assessment of skin sensitization, irritability and toxicity of bacteriocins and reuterin for possible topical applications”. Therefore, authors should explore if there is any research/concern regarding the potential toxicity or allergenicity of nisin in human.

Authors: The paper recommended by the reviewer is really novelty and interesting at all. In fact, we have included some comment in the “Introduction” section as follow:

“An interesting study provided important evidence that nisin Z could be also safely used topically as disinfectant at recommended concentrations without skin toxicity [20].”

At present, nisin was the bacteriocin approved by United States Food and Drug Administration for use as a food preservative. The safety and efficacy of nisin as a food preservative have resulted in its widespread use throughout the world.

4. This study showed that co-expressing nisin A and nisin Z resulted in a potentiation of activity against M. luteus only. It would be interesting to conduct additional antibacterial tests to evaluate the activity against other bacterial strains.

Authors: Good question. We agree with the reviewer that the evaluation activity against other bacterial strains would be interesting. However, we used the indicator microorganism Micrococcus luteus, which it is recommended as nisin-sensitive microorganisms in agar diffusion bioassay. We agree that we should evaluate additional bacterial strains. 

5. Please update the paper with latest references. There is only one reference from year 2022, while no reference from year 2023.

Authors: We agree with the reviewer. We have included latest references in the manuscript.

Dai Z, Han L, Li Z, Gu M, Xiao Z, Lu F. Combination of Chitosan, Tea Polyphenols, and Nisin on the Bacterial Inhibition and Quality Maintenance of Plant-Based Meat. Foods. 2022; 11(10):1524.

Maresca D, Mauriello G. Development of Antimicrobial Cellulose Nanofiber-Based Films Activated with Nisin for Food Packaging Applications. Foods. 2022; 11(19):3051. 

Wu M, Ma Y, Dou X, Zohaib Aslam M, Liu Y, Xia X, Yang S, Wang X, Qin X, Hirata T, Dong Q, Li Z. A review of potential antibacterial activities of nisin against Listeria monocytogenes: the combined use of nisin shows more advantages than single use. Food Res Int. 2023; 164:112363.

Chen H, Ji PC, Qi YH, Chen SJ, Wang CY, Yang YJ, Zhao XY, Zhou JW. Inactivation of Pseudomonas aeruginosa biofilms by thymoquinone in combination with nisin. Front Microbiol. 2023; 13:1029412.

Soltani S, Boutin Y, Couture F, Biron E, Subirade M, Fliss I. In vitro assessment of skin sensitization, irritability and toxicity of bacteriocins and reuterin for possible topical applications. Sci Rep. 2022; 12(1):4570.

Khazaei Monfared Y, Mahmoudian M, Cecone C, Caldera F, Zakeri-Milani P, Matencio A, Trotta F. Stabilization and Anticancer Enhancing Activity of the Peptide Nisin by Cyclodextrin-Based Nanosponges against Colon and Breast Cancer Cells. Polymers (Basel). 2022; 14(3):594.

Serrano I, Alhinho B, Cunha E, Tavares L, Trindade A, Oliveira M. Bacteriostatic and Antibiofilm Efficacy of a Nisin Z Solution against Co-Cultures of Staphylococcus aureus and Pseudomonas aeruginosa from Diabetic Foot Infections. Life (Basel). 2023; 13(2):504.

 

Reviewer #2: 

Nice presentation. Good and interesting piece of work. This is a very significant piece of work. This type of work must be published, tables and figure are well presented. Useful innovative work with industrial application.

Authors: We appreciated very much the opinion of the reviewer about the topic of our work.

---

## [Editor Report · Decision Letter 1]

27 Mar 2023

Bioengineering of a Lactococcus lactis subsp. lactis strain enhances nisin production and bioactivity.

PONE-D-23-01318R1

Dear Dr. Orlando Borras-Hidalgo,

We’re pleased to inform you that your manuscript has been judged scientifically suitable for publication and will be formally accepted for publication once it meets all outstanding technical requirements.

Kind regards,

Awatif Abid Al-Judaibi, PhD

Academic Editor

PLOS ONE

---

## [Editor Report · Acceptance letter]

31 Mar 2023

PONE-D-23-01318R1 

Bioengineering of a *Lactococcus lactis* subsp. lactis strain enhances nisin production and bioactivity. 

Dear Dr. Borras-Hidalgo:

I'm pleased to inform you that your manuscript has been deemed suitable for publication in PLOS ONE. Congratulations! Your manuscript is now with our production department. 

Kind regards, 

on behalf of

Professor Awatif Abid Al-Judaibi 

Academic Editor

PLOS ONE